# Investigation of Low-Pressure Sn-Passivated Cu-to-Cu Direct Bonding in 3D-Integration

**DOI:** 10.3390/ma15217783

**Published:** 2022-11-04

**Authors:** Po-Yu Kung, Wei-Lun Huang, Chin-Li Kao, Yung-Sheng Lin, Yun-Ching Hung, C. R. Kao

**Affiliations:** 1Department of Materials Science and Engineering, National Taiwan University, Taipei 10617, Taiwan; 2Product Characterization, Corporate R&D, Advanced Semiconductor Engineering (ASE) Group, Kaohsiung City 811, Taiwan

**Keywords:** 3D integration, Cu-to-Cu bonding, low pressure, compensate height difference

## Abstract

Cu-to-Cu direct bonding plays an important role in three-dimensional integrated circuits (3D IC). However, the bonding process always requires high temperature, high pressure, and a high degree of consistency in height. In this study, Sn is passivated over electroplated copper. Because Sn is a soft material and has a low melting point, a successful bond can be achieved under low temperature and low pressure (1 MPa) without any planarization process. In this experiment, Sn thickness, bonding temperature, and bonding pressure are variables. Three values of thicknesses of Sn, i.e., 1 μm, 800 nm, and 600 nm were used to calculate the minimum value of Sn thickness required to compensate for the height difference. Additionally, the bonding process was conducted at two temperatures, 220 °C and 250 °C, and their optimized parameters with required pressure were found. Moreover, the optimized parameters after the Cu planarization were also investigated, and it was observed that the bonding can succeed under severe conditions as well. Finally, transmission electron microscopy (TEM) was used to observe the adhesion property between different metals and intermetallic compounds (IMCs).

## 1. Introduction

Recently, the shrinking of transistors has met physical limitations, because extremely small transistors cause current leakage and damage electronic devices. Therefore, 3D integration has become a promising way to implement Moore’s law, as it has several advantages, such as high interconnection density, high performance, and small form factor [1,2]. There are two bonding methods in 3D IC: dielectric and metal bonding, and metal bonding plays a more important role because it determines the transmission signal and power.

In metal bonding, Cu-to-Cu direct bonding is the preferred method, owing to its low cost and good electrical conductivity [3,4]. However, the Cu-to-Cu bonding process is always conducted under high temperature and high pressure, which causes damage to electronic devices [5,6]. Moreover, the amount of Cu oxide on the surface is a crucial factor affecting the bonding quality [7]. Thus, several studies have passivated different inert materials, such as Au and Ag, over the Cu to lower the bonding temperature and pressure and to protect the Cu from oxidation [8,9,10,11]. Nonetheless, Cu-to-Cu direct bonding and metal-passivated Cu-to-Cu bonding both require high surface flatness [12], which is achieved by Chemical Mechanical Planarization (CMP) and grinding processes [13,14,15]. However, these are time-consuming and expensive processes. Therefore, a bonding process that could omit the CMP and grinding processes and can be conducted under low pressure and low temperature would be ideal. 

Due to the motivation mentioned above, in this study, Sn was passivated over Cu to enhance the bonding. Three bonding parameters, Sn thickness, bonding temperature, and bonding pressure, were investigated in the experiment. A schematic of Cu-to-Cu direct bonding with the Sn passivation layer is shown in Figure 1. Because Sn is soft, owing to its low melting point (232 °C) [16,17], a perfect bond can be achieved under low temperature and low pressure and without CMP. This could significantly reduce the process cost, process time, and avoid high thermal budgets and compressive stress. Experiments at two bonding temperatures were conducted in this study to find the optimal parameters. These temperatures were 220 °C (<Sn melting point (Tm)) and 250 °C (>Sn Tm). At 220 °C, Sn remains solid during the bonding; this could protect the other part of electronic device which also contains Sn from melting. At 250 °C, Sn becomes a liquid, so any height difference in the metals can be better compensated for, and higher surface roughness may be tolerated. Therefore, Sn of three thickness values (1 μm, 800 nm, and 600 nm) were utilized in this study to determine the minimum required thickness and the required bonding pressure. Additionally, the present paper will compare and discuss the bonding results of Sn passivated on Cu with CMP and Sn passivated on Cu without CMP.

## 2. Experimental Methods

A 50-nm Ti adhesion layer and a 300-nm Cu seed layer were sequentially sputtered onto a 4-inch silicon wafer. Subsequently, approximately 5 μm of Cu and different thicknesses of Sn were electroplated using a Cu and Sn electroplating solution from Sheng Hung Chemical Engineering Corporation in Taiwan. After electroplating, the wafer was sliced into small chips. To achieve good bonding, the chips were cleaned with acetone and 50 vol% HCl (36 wt%) followed by a DI water rinse to remove any organic materials or oxides on the sample surface. 

Afterward, chips were bonded by TCB (Thermal Compressive Bonding) under a 10^−2^ torr vacuum environment. The bonding profile is shown in Figure 2. First, two chips were pressed to avoid displacement. Then, at a heating rate of 40 °C/min, the target temperature was achieved (220 °C and 250 °C) and was held for 1 min. After completion of the bonding process, the pressure was removed and a N_2_ purge was used to cool down the sample to room temperature.

These samples were then mounted in epoxy and polished by SiC abrasive papers. To avoid the manual polishing of artifacts and surface impurities which could affect observations, an ion milling system (Hitachi IM4000Plus, Tokyo, Japan) with an Ar^+^ ion beam was applied after the manual polishing. Then, the bonding interface was analyzed by scanning electron microscopy (SEM, Hitachi SU5000, Tokyo, Japan) to examine the cross-sectional morphology. The chemical composition of Cu–Sn intermetallic compounds (IMCs) was measured by energy-dispersive X-ray spectrometry (EDX). Some nanovoids and adhesion properties between different layers could not be observed clearly under the SEM. Therefore, transmission electron microscopy (TEM, FEI Tecnai G2 F20, Hillsboro, OR, USA) was used to analyze the interface under extremely high magnification.

## 3. Results and Discussion

### 3.1. Bonding Parameter Optimization

Figure 3 shows the SEM image of different Sn thicknesses which were bonded at 220 °C and 250 °C under a pressure of 1 MPa for 1 min. As can be seen from Figure 3a,b, Sn bonded at both temperatures demonstrated good bonding quality, indicating that 1 μm of Sn was enough to compensate for the height difference and achieve perfect bonding at these two temperatures. It is well-known that Cu and Sn diffuse and form Cu_6_Sn_5_ and Cu_3_Sn; Cu_3_Sn is considered better than Cu_6_Sn_5_ due to its higher melting point and lower resistivity [18,19,20]. Although Cu_6_Sn_5_ would eventually become Cu_3_Sn due to the high working temperature of the electronic device, it was necessary to analyze the adhesion properties between the different metals and IMC layers while Cu_6_Sn_5_ was still present in order to ensure that the bonding strength was sufficient. This phenomenon was observed in the following SEM and TEM analysis. Sn becomes a liquid at 250 °C, and it reacted quickly with Cu to form IMCs. Therefore, after bonding, only Cu_6_Sn_5_ and Cu_3_Sn appeared at the interface, as shown in Figure 3a, but there was still some Sn remaining, as shown in Figure 3b. The bonding results when the Sn layer was reduced to 800 nm are shown in Figure 3c,d. All the IMCs became Cu_3_Sn at 250 °C, and the interface showed Cu_6_Sn_5_ and Cu_3_Sn with a low quantity of Sn at 220 °C. Furthermore, there were large gaps observed at the interface, which indicated that under these conditions, bonding did not go well, and this poor bonding might be a result of insufficient Sn. It has been reported that when two hard IMCs contact each other, the pressure is concentrated at the point of contact [21]. Thus, other points did not receive enough force, and this led to long gaps. For the 600 nm layer of Sn, because of the lower quantity of Sn, Figure 3e,f showed bigger gaps and holes at the interface. Moreover, at this thickness, no more Sn remains at 220 °C, and only Cu_6_Sn_5_ and Cu_3_Sn can be seen at the interface. Furthermore, all the IMCs were converted to Cu_3_Sn at 250 °C. 

It has been reported that higher pressure could lead to better bonding [22]. Therefore, the pressure was increased from 1 MPa to 2 MPa for those parameters for which bonding was poor (refer to Figure 3). Furthermore, Figure 4 shows the result when using 2 MPa for 800 nm Sn and 600 nm Sn at 220 °C and 250 °C, respectively. From Figure 4a, the condition of 800 nm Sn bonded at 250 °C indicated a good interface when using higher pressure, and all the IMCs were converted to Cu_3_Sn. Though most of the surfaces bonded well in Figure 4b, the holes were large. This could be caused by high surface roughness, because when Sn reacted with Cu to form smooth Cu_3_Sn and scallop-like Cu_6_Sn_5_, the depletion of Sn would make the surface rougher as shown in Figure 5 [23,24]. Thus, after two interfaces which had high surface roughness bonded with each other, the large holes finally appeared at the bonding interface. Specifically, 800 nm Sn is not thick enough to compensate for surface roughness of this magnitude. After all, the Sn remained solid at 220 °C, so it could not act like a liquid at 250 °C to flow and compensate for the height difference. For 600 nm Sn, small gaps still occurred at the interface under 2 MPa at 220 °C and 250 °C as shown in Figure 4c,d, which indicated that 2 MPa was not high enough to ensure perfect bonding. However, compared to Figure 3e,f, which are bonded under 1 MPa, the gaps were small. 

### 3.2. Porosity and Unbonded Interface Percentage 

Although there were four parameters that indicated good bonding results, their hole percentages and the size of the holes were different. Figure 6 shows the comparison of the hole percentage and unbonded region percentage among these four parameters. These two values were calculated using the following formulae:(1)$$\mathrm{Porosity}\;(\%)=\frac{\mathrm{Pores}\;\mathrm{Area}\;\left({\mu m}^{2}\right)}{\mathrm{Interface}\;\mathrm{IMCs}\;\mathrm{Area}\;\left({\mu m}^{2}\right)}$$
(2)$$\mathrm{Unbonded}\;\mathrm{Percentage}\;(\%)=\frac{\mathrm{Unbonded}\;\mathrm{Line}\;\mathrm{Length}\;\left(\mu m\right)}{\mathrm{Interface}\;\mathrm{Length}\;\left(\mu m\right)}$$

It was observed that the thickness of IMCs was not equal because their initial Sn thicknesses were different. Therefore, a variable called fixed interface IMCs area was set to calculate Equation (1) for each parameter. Moreover, the unbonded percentage was calculated with the unit of length in Equation (2).

In Figure 6, it can be seen that 1 μm Sn at 250 °C under 1 MPa was the best combination of parameters because it had the lowest porosity and the lowest unbonded percentage, approximately 0.9% and 13%, respectively. 1 μm Sn at a temperature of 220 °C under a pressure of 1 MPa and 800 nm Sn at a temperature of 250 °C under a pressure of 2 MPa showed similar results. Their porosities were around 1.5%, and unbonded percentages were approximately 20–25%. Finally, 800 nm Sn at 220 °C under 2 MPa had the largest porosity and unbonded percentage at 4.7% and 37.5%, respectively. Additionally, the porosity and unbonded percentage in the first three parameters showed a similar trend, as shown in Figure 6. Nevertheless, porosity increased more dramatically than the unbonded percentage for 800 nm Sn at 220 °C under 2 MPa because of the large holes caused by high surface roughness. The cause of this high surface roughness is discussed in the next paragraph.

### 3.3. Bonding after Grinding and CMP

In the previous section, the experiments were conducted without any planarization process. However, bonding parameter optimization after the copper planarization process is illustrated in this section. Figure 7 shows a 2D image of the electroplated Cu and Sn on electroplated copper before and after the planarization of the copper, obtained through Atomic Force Microscopy (AFM, Bruker Bioscope resolve). As shown in Figure 7a, electroplated Cu without grinding and CMP had much higher surface undulation than electroplated Cu with grinding and CMP (Figure 7b) because it did not undergo flattening. Previous research has demonstrated that electroplated Sn contains large grains [25], so the surface had high roughness after electroplating Sn. Consequently, under the AFM 2D image, grain morphologies of Sn were apparent, which is shown in Figure 7c,d. Table 1 shows the surface roughness (Rq) of each condition. The Rq value of electroplated copper was 22.1 nm, and the surface roughness was lowered to 1.32 nm after grinding and CMP. In addition, Sn on electroplated copper also showed a smaller Rq value with copper planarization than without copper planarization, which meant that the copper which was flatter could make the surface of electroplated Sn flatter as well. Table 2 shows the maximum height difference of the samples measured by the Alpha step (Surfcoder ET3000). It has been reported that electroplated copper had a ±10% height error [26,27], and the electroplated Cu possessed a 0.96 μm height difference. These data corroborated previous research, as the copper thickness was around 5–6 μm. After grinding and CMP, the height difference was reduced to 0.28 μm. Moreover, the Sn on electroplated Cu decreased from 0.84 μm to 0.42 μm.

Figure 8 shows the cross-sectional SEM bonding interface comparison of 800 nm Sn with and without the planarization process at 220 °C under 2 MPa for 1 min. Without grinding and CMP, many large holes were observed at the interface due to the high surface roughness, as shown in Figure 8a. However, after the planarization process, the bonding interface achieved perfect bonding without any apparent holes, as shown in Figure 8b. The porosity dropped from 4.8% to 0.3%, while the unbonded percentage decreased from 37.3% to 7.35%.

Furthermore, the flatter Cu not only made the holes smaller, but it also allowed the unbonded interface to change to a bonded interface under the same parameter. Figure 9a,a1 shows the bonding interface comparisons between 600 nm Sn with and without the planarization process at 220 °C under 2 MPa for 1 min. There was a long gap that appeared in the material without grinding and CMP (Figure 9a). However, after grinding and CMP, most of the surfaces connected, with some voids appearing at the interface, as shown in Figure 9a1. The porosity was 1.25% and the unbonded percentage was 16.9%. Also, Figure 9b,b1 shows a comparison between the bonding interface before and after the planarization process, both bonded at 250 °C under 1 MPa for 1 min with 800 nm Sn. Without grinding and CMP, the interface had an apparent gap, while the interface was well-connected after flattening. The porosity and the unbonded percentage of this well-connected interface were 1.21% and 22.7%, respectively. This improvement could be attributed to the flatter surface, which prevented the two hard IMCs from contacting each other first.

### 3.4. Summary of Bonding Parameters and Results

This section summarizes the bonding parameters and their qualities, which were discussed in the previous paragraph. Table 3 is the summary of the porosities and unbonded percentages of optimized parameters for achieving good bonding without any planarization process. A successful bonding under 1 MPa is only feasible in the case of 1 μm Sn thickness. When using 800 nm Sn thickness, 2 MPa pressure is required. Moreover, under the same Sn thickness and pressure, the sample prepared at 250 °C bonding temperature possesses fewer defects than the one prepared at 220 °C, because the melted Sn can easily flow and fill the holes. Moreover, although the pressure exerted on 1 μm Sn is lower than on 800 nm Sn, it still shows better bonding quality without any planarization process.

Table 4 shows that the optimized bonding parameter changes after copper planarization when compared to no planarization. The drastic decrease in the surface roughness and maximum height difference leads to a better bonding quality and allows the bonding to succeed even under suboptimal conditions. The porosity and unbonded percentage show a drastic decrease in the 800 nm thick Sn, bonded at 220 °C under 2 MPa for 1 min. Additionally, the pressure can decrease to 1 MPa in the 800 nm thick Sn, bonded at 250 °C, and Sn thickness could be reduced to 600 nm under 2 MPa at 250 °C after grinding and CMP.

### 3.5. TEM Analysis

There was only Cu_3_Sn at the interface for some parameters, while the other parameters had Cu_6_Sn_5_ and remaining Sn after the bonding. However, these Cu_6_Sn_5_ and Sn were eventually converted to Cu_3_Sn due to the high working temperature. It is still important to measure the adhesion properties between different layers no matter what type of IMCs were at the interface, and there might be some defects and voids which could not be observed clearly under the SEM. Therefore, TEM was required to analyze the bonding quality between different IMCs and metal layers under higher magnification. In addition, a diffraction pattern was used to perform phase identification. Figure 10a shows the bright-field TEM image of the interface bonded at 250 °C under 1 MPa for 1 min with 1 μm Sn thickness. Under this condition, the interface had a Cu/Cu_6_Sn_5_/Cu_3_Sn/Cu_6_Sn_5_/Cu structure. Figure 10b,c shows the selected area diffraction patterns (SADPs) of the Cu_3_Sn and Cu_6_Sn_5_, respectively, corresponding to Figure 10a. Cu_3_Sn is the columnar grain, which was in accord with the previous study [28]. Additionally, there were no voids or cracks at either the Cu–Cu_3_Sn interface or the Cu_6_Sn_5_–Cu_3_Sn interface. This phenomenon could indicate that the interface had good adhesion. The stress concentration caused by voids or cracks would not happen in this system. Moreover, the study indicated that the Sn–Cu_6_Sn_5_ interface is the main factor that affected the strength of the Cu–Sn system [29]. However, in this experiment, only a small amount of Sn remained at the interface. Furthermore, the remaining Sn was scattered in the Cu_6_Sn_5_ layer, and the Sn was depleted in most of the conditions after the bonding process. Therefore, the strength of the joints could not be determined by the Sn–Cu_6_Sn_5_ interface. Moreover, the previous studies revealed that the Cu/Cu_6_Sn_5_/Cu_3_Sn/Cu_6_Sn_5_/Cu structure had good strength [30,31], so the joint with Cu_6_Sn_5_ that obtained in this experiment would not be fragile.

Figure 11a shows the TEM bright-field image after all the Cu_6_Sn_5_ was converted to Cu_3_Sn with 800 nm thick Sn bonded at a temperature of 250 °C under 2 MPa pressure for 1 min. While in this state, grain growth stopped; the Cu_3_Sn grain layers on both sides came in contact with each other but did not merge together, which was in accord with the previous study [28]. Figure 11b shows the SADP of the Cu_3_Sn grain shown in Figure 11a. It was reported that Cu/Cu_3_Sn/Cu had over 44 Mpa of shear strength [32]. No defects occurred at the Cu–Cu_3_Sn interface, so the joint would also be strong when all the IMCs converted to Cu_3_Sn.

## 4. Conclusions

To develop a Cu-to-Cu direct bonding with low temperature and low pressure and without any planarization process, passivating a Sn layer over the copper material is a promising method. Despite using the surface with a roughness of 22.1 nm Rq and 0.96 μm height difference, the bonding result is good under the conditions mentioned above. These bonding parameters were optimized, and the bonding qualities were investigated using SEM and TEM. The following conclusions can be drawn:As shown in Table 3, there are three parameter changes in this study: Sn thickness, bonding temperature, and bonding pressure. From the results, higher Sn thickness (1 μm) and higher temperature (250 °C) are recommended to achieve better bonding. Moreover, if a lower Sn thickness (800 nm) is used, 2 MPa pressure can still achieve a successful bond. However, the bonding quality is not as good as the parameters of 1 μm Sn under 1 MPa pressure at both 250 °C and 220 °C. Besides, 600 nm Sn is not thick enough to bond well without the planarization process in spite of the 250 °C temperature and 2 MPa pressure.Though the high surface flatness is not needed in the bonding after passivating soft Sn over Cu, the surface roughness of Cu will still affect the bonding quality. The comparison between planarization and no planarization of required bonding parameters and the bonding results are shown in Table 4.As per the TEM analysis, there are no cracks or voids which occur at the Cu–Cu_3_Sn, Cu_3_Sn–Cu_6_Sn_5_, or Cu_3_Sn–Cu_3_Sn interfaces. This implies that there is good adhesion between different layers regardless of whether Cu_6_Sn_5_ is present at the interface. Therefore, the strength of this Cu–Sn joint would be sufficiently strong, because no defects occur between different layers. Thus, the defects at the bonding interface might be the dominant factor which affects the joint strength.

Finally, by passivating Sn over Cu, apart from the advantage of a low-temperature and low-pressure bonding process, the most attractive quality is that it does not require extremely low surface roughness. Because CMP is currently a time-consuming and expensive procedure and is always required before bonding in 3D IC, such as hybrid bonding, this research will provide a promising method to enhance the bonding process without grinding or CMP. Additionally, the whole bonding process can be accomplished with a cheaper and faster procedure.

## Figures and Tables

**Figure 1 materials-15-07783-f001:**
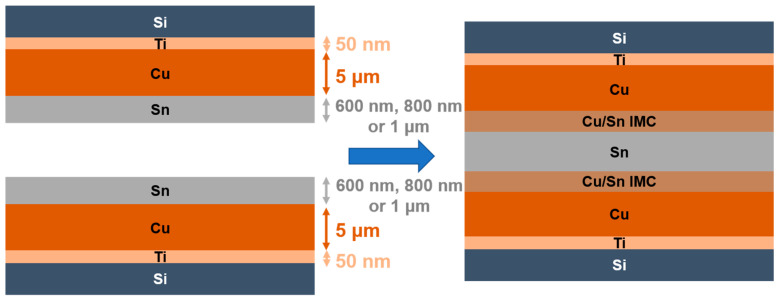
Schematic of the Cu-to-Cu direct bonding using Sn as the passivation layer.

**Figure 2 materials-15-07783-f002:**
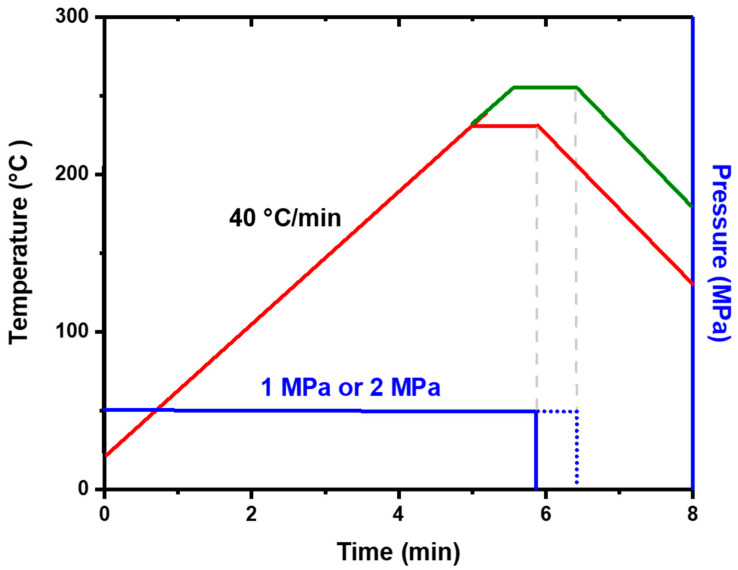
Pressure and temperature profile of the bonding process.

**Figure 3 materials-15-07783-f003:**
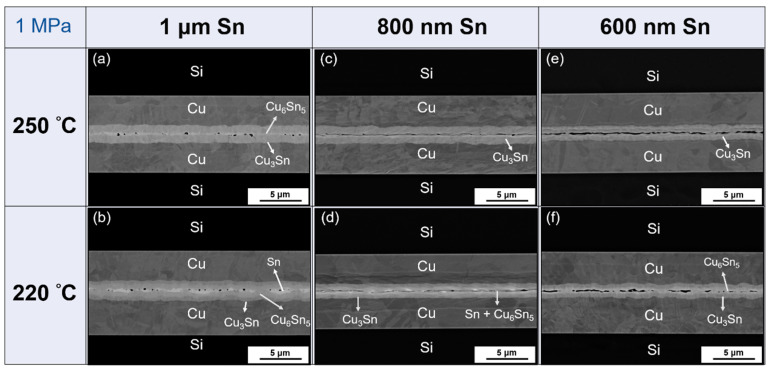
Cross-sectional SEM images of the bonding interface under 1 MPa for 1 min. (**a**) 250 °C, 1 μm Sn; (**b**) 220 °C, 1 μm Sn; (**c**) 250 °C, 800 nm Sn; (**d**) 220 °C, 800 nm Sn; (**e**) 250 °C, 600 nm Sn; (**f**) 220 °C, 600 nm Sn.

**Figure 4 materials-15-07783-f004:**
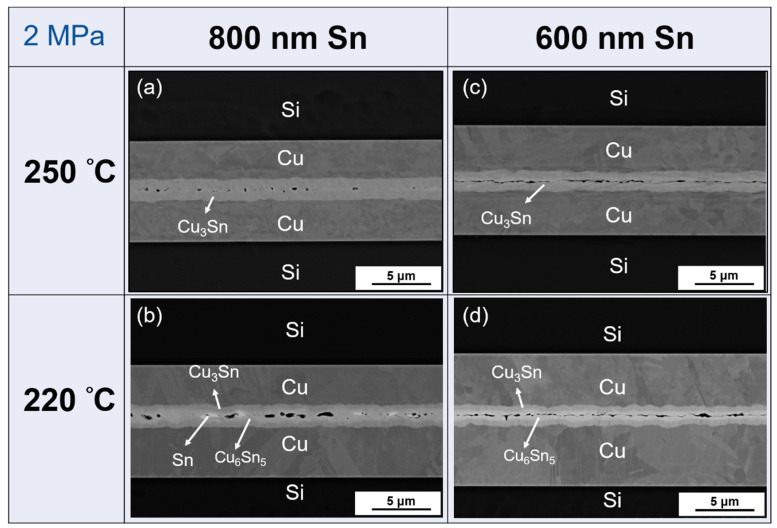
Cross-sectional SEM image of the bonding interface under 2 MPa for 1 min. (**a**) 250 °C, 800 nm Sn; (**b**) 220 °C, 800 nm Sn; (**c**) 250 °C, 600 nm Sn; (**d**) 220 °C, 600 nm Sn.

**Figure 5 materials-15-07783-f005:**
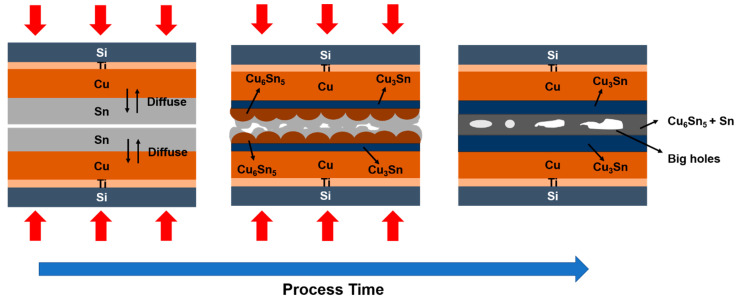
Schematic of IMC formation and microstructure evolution of the Cu–Sn bonding when Sn is not thick enough.

**Figure 6 materials-15-07783-f006:**
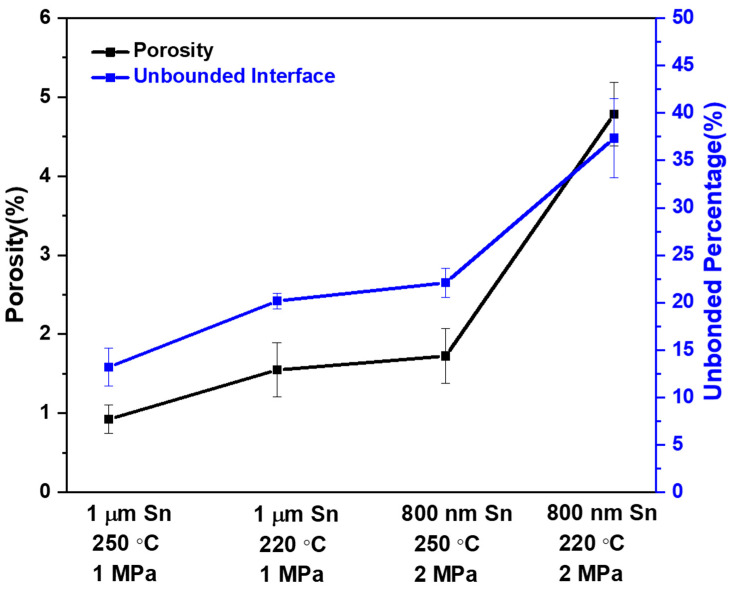
Comparison of porosity and unbonded percentage between four different parameters.

**Figure 7 materials-15-07783-f007:**
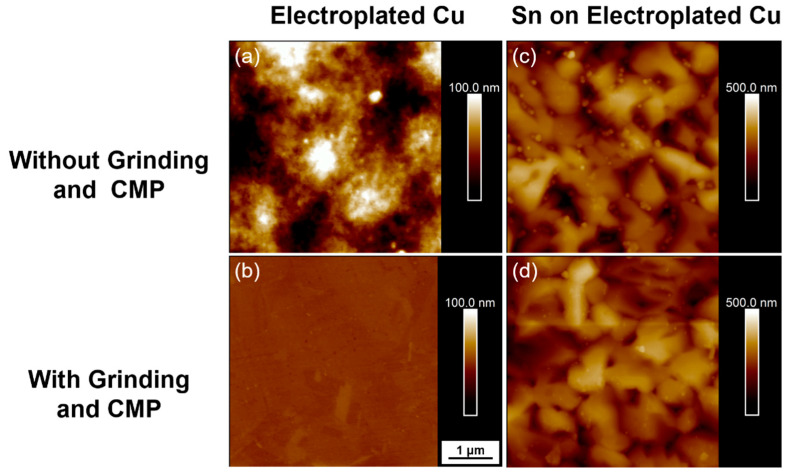
AFM 2D image of (**a**) Electroplated Cu without grinding and CMP; (**b**) Electroplated Cu with grinding and CMP; (**c**) Sn on electroplated Cu without grinding and CMP; (**d**) Sn on electroplated Cu with grinding and CMP.

**Figure 8 materials-15-07783-f008:**
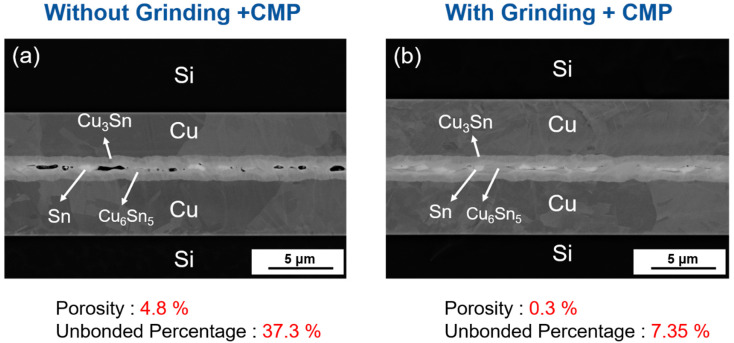
Comparison between SEM bonding interface (**a**) without and (**b**) with planarization process with 800 nm Sn at 220 °C under 2 MPa for 1 min.

**Figure 9 materials-15-07783-f009:**
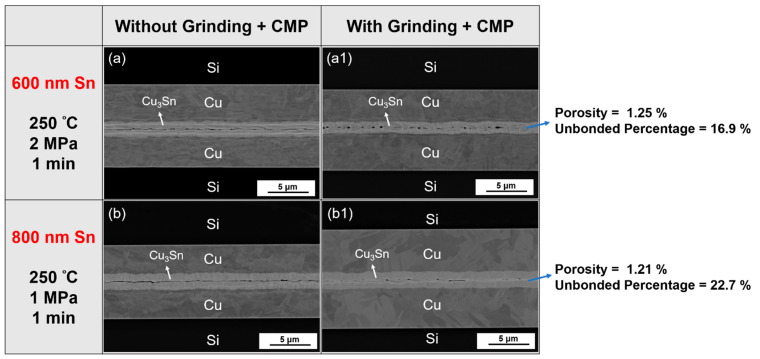
Comparison between (**a**,**b**) without grinding and CMP, and (**a1**,**b1**) with grinding and CMP under two bonding parameters.

**Figure 10 materials-15-07783-f010:**
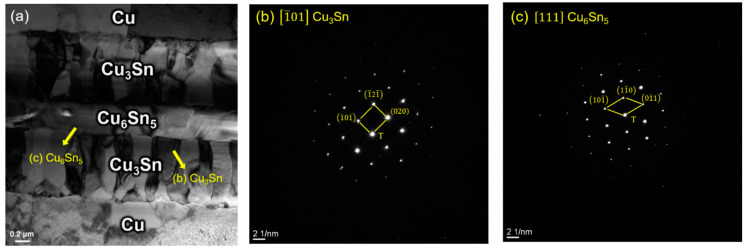
(**a**) Bright Field TEM of Cu/Cu_6_Sn_5_/Cu_3_Sn/Cu_6_Sn_5_/Cu structure under the condition of 1 μm Sn, bonded at 250 °C under 1 MPa for 1 min; (**b**) SADPs of Cu_3_Sn; (**c**) SADPs of Cu_6_Sn_5_.

**Figure 11 materials-15-07783-f011:**
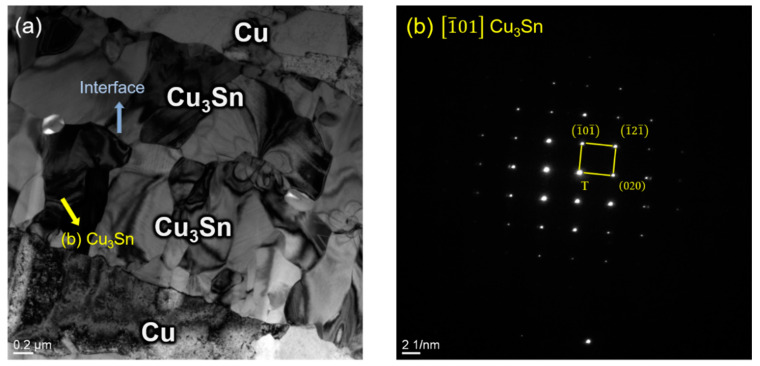
(**a**) Bright-field TEM of Cu–Cu_3_Sn structure; (**b**) SADPs of Cu_3_Sn.

**Table 1 materials-15-07783-t001:** AFM measurement of four different parameters.

Rq	Electroplated Cu	Sn on Electroplated Cu
Without Grinding + CMP	22.1 nm	61.1 nm
With Grinding + CMP	1.32 nm	56.2 nm

**Table 2 materials-15-07783-t002:** Maximum height difference of four different parameters.

Maximum Height Difference	Electroplated Cu	Sn on Electroplated Cu
Without Grinding + CMP	0.96 μm	0.84 μm
With Grinding + CMP	0.28 μm	0.42 μm

**Table 3 materials-15-07783-t003:** Summary of the optimized parameters without grinding or CMP.

Sn Thickness	Temperature	Pressure	Time	Porosity	Unbonded Percentage
1 μm	250 °C	1 MPa	1 min	0.9%	13.2%
	220 °C	1 MPa	1 min	1.5%	20.2%
800 nm	250 °C	2 MPa	1 min	1.7%	22.1%
	220 °C	2 MPa	1 min	4.8%	37.3%

Without Grinding or CMP.

**Table 4 materials-15-07783-t004:** Summary of the optimized parameters with grinding and CMP (red characters show the differences, when compared to no grinding and CMP).

Sn Thickness	Temperature	Pressure	Time	Porosity	Unbonded Percentage
800 nm	220 °C	2 MPa	1 min	4.8% → 0.3%	37.3% → 7.35%
800 nm	250 °C	2 MPa → 1 MPa	1 min	1.21%	22.7%
800 nm → 600 nm	250 °C	2 MPa	1 min	1.25%	16.9%

With Grinding + CMP.

## Data Availability

Not applicable.

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
