# Peer review of "Investigation of Low-Pressure Sn-Passivated Cu-to-Cu Direct Bonding in 3D-Integration"

_materials, 2022, doi:10.3390/ma15217783_

Round 1

Reviewer 1 Report

1. The study explores the impact of Sn layer thickness, bonding temperature and pressure on the quality of Sn-passivated Cu-to-Cu direct bonding.

2. I consider the topic relevant, and the novelty is related not particularly to the methods applied, but to the combinations of process parameters used.

3. It contributes with an observation of impact of some relevant process parameters

4. The observation is performed at SEM and TEM microscopy, but no mechanical test is then performed. I would consider a quantification of the quality through testing in further research.

5. The conclusions are consisted with what presented and discussed.

6. The references are relevant, and I did not incur in inappropriate self-citation.

7. Tables and figures are of good quality.

8. Start of Page 11, point 1 presents a stroke on the number.

Author Response

The article is full of important content. I enjoyed reading it. While I am recommending publication, I think that it need of several major revision that can improve quality and readability of the text:

  1. The study explores the impact of Sn layer thickness, bonding temperature and pressure on the quality of Sn-passivated Cu-to-Cu direct bonding.
  2. I consider the topic relevant, and the novelty is related not particularly to the methods applied, but to the combinations of process parameters used.
  3. It contributes with an observation of impact of some relevant process parameters
  4. The observation is performed at SEM and TEM microscopy, but no mechanical test is then performed. I would consider a quantification of the quality through testing in further research.
  5. The conclusions are consisted with what presented and discussed.
  6. The references are relevant, and I did not incur in inappropriate self-citation.
  7. Tables and figures are of good quality.
  8. Start of Page 11, point 1 presents a stroke on the number.

Response: Thank you for your comments. We so appreciate your time and consideration. The stroke on the number has been deleted. And for the mechanical test, because there are no apparent fractures or big holes that happened under the SEM and TEM, we believed the joints possess good mechanical strength. But we will consider doing the mechanical strength in the future so that the research will be more complete.

Reviewer 2 Report

In this manuscript, a low-pressure Cu-Cu directed bonding techniques was developed. Overall, the work is of interest to the readers of the journal and well written. However, there are few comments that the authors should address thus minor revision is requested. Please refer to the following comments:

1.      The introduction appears a bit brisk and abrupt in its construction.

2.      The manuscript only examines the characterization of the bonding and the processing route involved, but the mechanical testing of the joint would seem like a suitable addition to this work. The authors can state their point of view.

3.      The introduction of a new Table in conclusions is not the most ideal way. The Tables 3 and 4 can be moved into the results and their importance on this work can be summarized in conclusions.

Author Response

In this manuscript, a low-pressure Cu-Cu directed bonding techniques was developed. Overall, the work is of interest to the readers of the journal and well written. However, there are few comments that the authors should address thus minor revision is requested. Please refer to the following comments:

  1. The introduction appears a bit brisk and abrupt in its construction.

Response: Thank you for your comments. I have added the process parameters that we are investigating in the introduction and revised the introduction to make it more well-organized.

  1. The manuscript only examines the characterization of the bonding and the processing route involved, but the mechanical testing of the joint would seem like a suitable addition to this work. The authors can state their point of view.

Response: Thank you for your comments. The most fragile parts in the joint usually happened at the interface of different layers, such as Cu/Cu6Sn5, or Cu6Sn5/Cu3Sn, or the bonding interface where holes appeared. First of all, the holes percentage at the bonding interface is not high, so it will not be a problem that affects the joint strength. Secondly, from the SEM and TEM, there are no voids that occur between these interfaces, indicating that the stress concentration may not happen at the interface between different layers. Also, from reference [1,2], it has shown that Cu/Cu6Sn5/Cu3Sn/Cu6Sn5/Cu structure possessed good strength. Moreover, all of the IMCs will convert to Cu3Sn, which possesses high mechanical strength among all the Cu/Sn IMC, after a high-temperature working environment. As a result, we believe that the joint will have strong mechanical strength.

[1] Kao, C.-W., P.-Y. Kung, C.-C. Chang, W.-C. Huang, F.-L. Chang, and C. Kao, Highly Robust Ti Adhesion Layer during Terminal Reaction in Micro-Bumps. Materials, 2022. 15(12): p. 4297.

[2] Yang, C., F. Song, and S.R. Lee. Effect of interfacial strength between Cu 6 Sn 5 and Cu 3 Sn intermetallics on the brittle fracture failure of lead-free solder joints with OSP pad finish. in 2011 IEEE 61st Electronic Components and Technology Conference (ECTC). 2011. IEEE.

  1. The introduction of a new Table in conclusions is not the most ideal way. The Tables 3 and 4 can be moved into the results and their importance on this work can be summarized in conclusions.

Response: Thank you for your comments. Tables 3 and 4 have been moved into the results, and we have added other conclusions.

The new conclusions we made in the articles:

  1. As shown in Table 3, there are three parameter changes in this research, Sn thickness, bonding temperature, and bonding pressure respectively. From the results, higher Sn thickness (1 μm) and higher temperature (250 °C) are recommended to be used to achieve better bonding. Moreover, if a lower Sn thickness (800 nm) is used, 2 MPa pressure can still make a successful bonding. However, the bonding quality is not as good as the parameters of 1 μm Sn with 1 MPa pressure under both 250 °C and 220 °C. Besides, 600 nm Sn is not thick enough to bond well without the planarization process in spite of the 250 °C temperature and 2 MPa pressure.

  1. Though the high surface flatness is not needed in the bonding after passivating soft Sn over Cu, the surface roughness of Cu will still affect the bonding quality. The comparison between with planarization process and without the planarization process of required bonding parameters and the bonding results are shown in Table

Reviewer 3 Report

The authors have written this paper very well. All the images and schematics are clear, sufficiently large and easily readable. There are very minor format and language corrections needed as the paper is very well written. 

This work is of great interest to semiconductor industry and the parameters investigated by authors will help engineers optimize Cu-Cu bonding process. One recommendation for authors is to clearly specify the process parameters they are investigating, in the abstract and at the end of introduction section. It wasn't directly clear in abstract, which parameters are under investigation.

Finally, adding a paragraph describing the significance, application and benefits of this research will enrich the value of this paper.

Author Response

The authors have written this paper very well. All the images and schematics are clear, sufficiently large and easily readable. There are very minor format and language corrections needed as the paper is very well written. 

Response: Thank you for your comments. We have revised the wrong formats, the stroke on the number in the conclusion has been deleted.

This work is of great interest to semiconductor industry and the parameters investigated by authors will help engineers optimize Cu-Cu bonding process. One recommendation for authors is to clearly specify the process parameters they are investigating, in the abstract and at the end of introduction section. It wasn't directly clear in abstract, which parameters are under investigation.

Response: Thank you for your comments. I have mentioned the parameters I used in the abstract and at the end of introduction section.

Finally, adding a paragraph describing the significance, application and benefits of this research will enrich the value of this paper.

Response: Thank you for your comments. We have added the significance and benefits of this research in the end of the conclusion.

The new paragraph that I added in the end of the conclusion:

Finally, by passivating Sn over Cu, apart from the advantage of low-bonding temperature and low-bonding pressure process, the most attractive feature is that it does not require extremely low surface roughness. Since nowadays CMP, which is a time-consuming and expensive procedure, is always required before the bonding in 3D IC, such as hybrid bonding, this research will provide a promising method to enhance the bonding without the grinding and CMP process. And the whole bonding process will be finished in a cheaper and faster procedure.
